# *FvHsfB1a* Gene Improves Thermotolerance in Transgenic *Arabidopsis*

**DOI:** 10.3390/plants14152392

**Published:** 2025-08-02

**Authors:** Qian Cao, Tingting Mao, Kebang Yang, Hanxiu Xie, Shan Li, Hao Xue

**Affiliations:** 1School of Horticulture, Anhui Agricultural University, Hefei 230036, China; 16634254430@163.com (Q.C.); 15156002326@163.com (T.M.); 19156213306@163.com (H.X.); 19960653711@163.com (S.L.); 2Shanghai Agricultural Technology Extension Service Center, Shanghai 201103, China; 15721351226@163.com

**Keywords:** strawberry, *FvHsfB1a*, *FvWRKY75*, heat stress, *Arabidopsis thaliana*

## Abstract

Heat stress transcription factor (Hsf) families play important roles in abiotic stress responses. However, previous studies reported that *HsfBs* genes may play diverse roles in response to heat stress. Here, we conducted functional analysis on a woodland strawberry Class B Hsf gene, *FvHsfB1a*, to improve thermotolerance. The structure of FvHsfB1a contains a typical Hsf domain for DNA binding at the N-terminus, and FvHsfB1a belongs to the B1 family of Hsfs. The FvHsfB1a protein was localized in the nucleus. The *FvHsfB1a* gene was expressed in various strawberry tissues and highly induced by heat treatment. Under heat stress conditions, ectopic expression of *FvHsfB1a* in *Arabidopsis* improves thermotolerance, with higher germination and survival rates, a longer primary root length, higher proline and chlorophyll contents, lower malonaldehyde (MDA) and O^2−^ contents, better enzyme activities, and greater expression of heat-responsive and stress-related genes compared to WT. FvWRKY75 activates the promoter of the *FvHsfB1a* gene through recognizing the W-box element. Similarly, *FvWRKY75*-OE lines also displayed a heat-tolerant phenotype, exhibiting more proline and chlorophyll contents, lower MDA and O^2−^ contents, and higher enzyme activities under heat stress. Taken together, our study indicates that *FvHsfB1a* is a positive regulator of heat stress.

## 1. Introduction

The growth and development of plants require suitable environmental conditions, generally including temperature, light, water, and soil nutrients [1]. Adverse environmental factors, especially heat stress, cause several morphological and physiological changes in plants. This includes protein denaturation, plasma membrane disruption, and reactive oxygen species (ROS) accumulation, and can lead to plant death in severe cases [2,3,4,5]. Plants have a series of complex physiological mechanisms and molecular regulatory networks to minimize the damage caused by the external environmental stresses [6,7].

The heat shock transcription factors (Hsfs) family is generally known to play a central role in regulating abiotic stress response [8,9,10]. Based on the conserved Hsf DNA-binding domain (DBD) and oligomerization domain (OD) regions, Hsfs can be divided into three classes in higher plants, namely HsfA, HsfB, and HsfC [11,12]. In a previous study, HsfAs were found to be master regulators of thermotolerance by regulating the expression of downstream genes in plants [13,14,15,16]. In contrast to HsfAs, HsfBs have no transcriptional activity, because they lack a C-terminal activator domain (CTAD) [17]. Previous studies reported that *HsfBs* are induced by heat stress and involved in the regulation of thermotolerance in many plant species [18,19]. For example, *AtHsfB1* and *AtHsfB2b* act as suppressors of the expression of *Hsfs* and several heat shock protein genes but are necessary for acquiring thermotolerance [20]; *PpHsf5* in peaches and *HsfB1* in grapes were reported to play a positive role in thermotolerance [21,22]. In addition, some *HsfBs* genes, such as *OsHsfB2b* from rice, *CarHsfB2* from chickpeas, *GmHsfB2b* from soybeans, and *ZmHsf08* (Class B gene) from maize, play important roles in drought and salt tolerance responses [23,24,25,26]. For HsfCs members, it is demonstrated that *HsfC2a* from wheat and *HsfC1b* from tall fescue regulate the heat-responsive genes to increase heat tolerance, and *VaHsfC1* from grapes is involved in multiple abiotic stresses [27,28,29,30].

The strawberry is cultivated widely in the world and is threatened by a range of adverse environmental factors. With global warming, the increasing number of extreme temperature events has severely affected the growth, development, productivity, and quality of strawberry, posing a considerable challenge to modern agricultural farming [31,32]. Seventeen *Hsf* genes have been identified in the diploid woodland strawberry (*Fragaria vesca*), and different *FvHsfs* have distinct responses to various biotic and abiotic stresses at the transcriptional level [33]. However, the molecular mechanisms of the *FvHsfs* genes involved in abiotic stress remain unclear. Herein, we cloned the *FvHsfB1a* gene from ‘Ruegen’ strawberry leaves and found that *FvHsfB1a* was induced by heat stress treatment. We then explored its function in heat stress response. The overexpression of the *FvHsfB1a* gene enhances heat stress resistance. Compared with WT plants, heat-responsive and stress-related genes in *FvHsfB1a*-OE lines were rapidly and strongly induced by heat stress treatment. Furthermore, the *FvHsfB1a* gene was directly regulated by the FvWRKY75 TF. The overexpression of the *FvWRKY75* gene in *Arabidopsis* also improved heat tolerance. Overall, we have provided evidence that *FvHsfB1a* may play an important role in improving strawberry thermotolerance.

## 2. Results

### 2.1. Isolation and Characterization of FvHsfB1a

The *FvHsfB1a* gene was isolated from ‘Ruegen’ leaves and the CDS sequence of *FvHsfB1a* was 873 bp, encoding a protein containing 290 amino acids (Figure 1A). The predicted molecular weight and isoelectric point of FvHsfB1a protein were predicted to be 32.01 kDa and 5.96 M. The structure of FvHsfB1a contained a typical HSF domain for DNA binding at the N-terminus (Figure 1A). Phylogenetic analysis showed that the sequence of FvHsfB1a was the closest homolog to RcHSF24 (class B1) (Figure 1B).

The subcellular localization of FvHsfB1a protein was detected via green fluorescent protein (GFP) and red fluorescent protein (RFP) as a nuclear location marker. As shown in Figure 1C, the recombinant pCAMBIA1302-FvHsfB1a-GFP was localized only in the nucleus.

### 2.2. Expression Profiles of FvHsfB1a in Diploid Strawberry

The tissue-specific expression patterns of *FvHsfB1a* were measured in the diploid strawberry ‘Ruegen’. The *FvHsfB1a* gene was differentially expressed in different tissues, with the lowest expression in leaves and the highest expression in the stem (Figure 2A). Then, we investigated the expression profiles of the *FvHsfB1a* gene by using mature strawberry leaves under abiotic stress conditions. Under heat stress treatment, *FvHsfB1a* gene expression was significantly induced and reached a peak with a 43.7-fold increase after 24 h of treatment (Figure 2B).

### 2.3. FvHsfB1a Improves Thermotolerance in Arabidopsis

To examine the function of *FvHsfB1a* under heat stress, we generated *FvHsfB1a* transgenic *Arabidopsis* plants, and two *FvHsfB1a*-OE T3 homozygous lines were selected by using PCR and qRT-PCR for future tests (Appendix A). As shown in Figure 3, no significant differences in seed germination and seedling growth were observed between the WT and OE lines under normal conditions. After treatment with heat stress for 7 d, the WT seeds exhibited the lowest germination rates, especially under the 42 °C treatment (Figure 3A,B). In contrast, the *FvHsfB1a*-OE seeds had higher germination rates in 37 °C and 42 °C treatments for 7 d compared to WT (Figure 3A,B). Meanwhile, the performance of *FvHsfB1a*-OE and WT *Arabidopsis* seedlings was evaluated after 45 °C treatment for 3 h. Compared with WT, there were better performances, higher survival rates, and longer root lengths in *FvHsfB1a*-OE seedlings observed (Figure 3C,D). Additionally, the heat tolerance of WT and *FvHsfB1a*-OE mature plants was also detected. Compared with WT, *FvHsfB1a*-OE lines showed a lower degree of leaf wilting, higher chlorophyll and proline contents, lower MDA and O_2_^−^ contents, and better superoxide (SOD) and peroxidase (POD) activities after heat stress treatment (Figure 4A–G). These results suggest that the *FvHsfB1a* gene could improve thermotolerance in *Arabidopsis*.

### 2.4. Overexpression of FvHsfB1a Regulates the Expression of Stress-Related Genes

To further investigate the regulation of *FvHsfB1a* in response to heat stress, the expression profiles of several heat-responsive and stress-related genes were detected by using qRT-PCR in the WT and *FvHsfB1a*-OE lines. After treatment with heat stress, the expression levels of heat-responsive genes (*AtHSP70b* and *AtHSP101*) and stress-related genes (*AtP5CS1*, *AtSOD*, and *AtPOD*) were significantly upregulated in *FvHsfB1a*-OE lines as compared with the WT (Figure 5).

### 2.5. FvWRKY75 Can Active the Promoter of FvHsfB1a Gene

Bioinformatics analysis showed that the promoter of the *FvHsfB1a* gene contains several WRKY binding elements (Figure 6A). Based on previous findings, FvWRKY70 and FvWRKY75 were selected for a yeast one-hybrid assay to determine their relationship with *FvHsfB1a*. As shown in Figure 6B,C, the FvWRKY75 directly bound to the *FvHsfB1a* promoter and influenced its expression in the yeast one-hybrid and LUC assays.

### 2.6. FvWRKY75 Enhances Thermotolerance in Arabidopsis

Previously, we cloned the *FvWRKY75* gene and generated two *FvWRKY75*-OE transgenic *Arabidopsis* plants [34]. In this study, we examined the role of the *FvWRKY75* gene in the response to high-temperature stress. During heat treatment, the *FvWRKY75*-OE transgenic plants exhibited greater insensitivity to heat stress with a better phenotype compared to the WT (Figure 7A). Compared with the WT, lower MDA and O_2_^−^ contents, higher chlorophyll and proline contents, and better peroxidase activity were observed in the *FvWRKY75*-OE plants (Figure 7B–G), which suggested that *FvWRKY75* is also a positive regulator of heat tolerance.

## 3. Discussion

Heat stress severely affects the growth, development, production, and yields of most agricultural crops [35,36,37,38]. The Hsf family has been well-identified in many plant species and functions as a key regulatory component involved in various stress responses [10,12,39]. In strawberries, 17 *Hsfs* were identified based on the wild diploid woodland strawberry (*Fragaria vesca*) [33]. Here, we cloned an Hsf TF gene, *FvHsfB1a*, from ‘Ruegen’ leaves, and explored its characterization. The amino acid sequence of FvHsfB1a showed all the characteristics of Class B HSFs, and the sequence of FvHsfB1a showed high homology with that of RcHsf24, which belongs to Class B1 Hsfs (Figure 1A,B). The FvHsfB1a protein was localized only in the nucleus (Figure 1C,D). Some Hsf B TF genes, including *OsHSF2b*, *CarHSFB2,* and *ZmHsf08,* were upregulated under abiotic stresses and proved to play a role in heat stress response [23,24,26]. In the current study, we found that the *FvHsfB1a* gene was induced by heat stress (Figure 2), which suggested that the *FvHsfB1a* gene may be involved in the heat stress response in strawberry.

Previous studies reported that *HsfBs* genes may play diverse roles in abiotic stress responses. For example, the overexpression of *VdHSFB1* and *VvHSFB1* in grapes improved heat tolerance, and the overexpression of *CarHSFB2* and *GmHsfB2b* improved resistance to drought and salt stress in transgenic plants [22,23,25]. However, *OsHsfB2b* and *ZmHsf08* (a member of *HSFB1*) negatively regulated drought and salt tolerance in rice and maize [24,26]. To better understand the function of the *FvHsfB1a* gene in strawberry, two *FvHsfB1a*-OE *Arabidopsis* lines were generated and used for future tests (Appendix A). After heat stress treatment, *FvHsfB1a*-OE lines showed increased insensitivity to high temperature, longer root length, higher germination rates, and higher survival rates than the WT (Figure 3E,F), which suggested that *FvHsfB1a* may regulate the growth of roots in response to heat stress. Under stress conditions, chlorophyll degradation and membrane lipid peroxidation occur in plants, resulting in a decrease in chlorophyll content and an increase in ROS (O^2−^ and H_2_O_2_) and MDA content. Therefore, chlorophyll, MDA, O^2−^, and H_2_O_2_ contents are commonly used to assess the severity of stress damage in plants [40,41,42]. In addition, antioxidant enzymes, such as POD, CAT, and APX, play important roles in scavenging oxidative damage caused by ROS accumulation [43,44]. In this study, it was noticed that *FvHsfB1a*-OE lines had a lower MDA content and higher chlorophyll and proline contents than WT after stress treatment; meanwhile, a lower O_2_^−^ content and higher SOD and POD activities were presented for the former rather than the latter (Figure 4), which suggested that *FvHsfB1a* could attenuate heat stress damage to plants by modulating ROS scavenging and increasing antioxidant enzyme activity.

Heat shock proteins (HSPs) were known as the target genes of HSFs and play important roles in heat stress regulation [45,46]. *HSP70* and *HSP101* are induced by heat stress and confer improved heat tolerance in *Arabidopsis* [47,48]. In the current study, *AtHSP70b* and *AtHSP101* were significantly induced in *FvHsfB1a*-OE lines compared to in WT plants under heat stress (Figure 5A,B). This suggests that *FvHsfB1a* could influence the expression of these *HSP* genes to improve thermotolerance in transgenic plants. The *P5CS* gene participated in the synthesis of proline, which contributes to the positive regulation of plant dehydration and osmotic stress [49]. The upregulation of the *AtP5CS1* gene in the *FvHsfB1a*-OE lines under heat treatment may contribute to the accumulation of proline, which can reduce the injury caused by stress (Figure 5C). Many studies reported the connection between antioxidant enzyme genes and Hsfs in scavenging ROS [50,51]. In the current study, the stress-related genes (*AtSOD* and *AtPOD*) showed higher expression levels in transgenic plants under heat stress conditions (Figure 5D,E). All of the results suggest that the overexpression of the *FvHsfB1a* gene improves thermotolerance by upregulating the expression of heat-responsive genes (*AtHSP70b* and *AtHSP101*) and stress-related genes (*AtP5CS1*, *AtSOD,* and *AtPOD*).

In this study, the Y1 H and LUC assays showed that the FvWRKY75 transcription factor directly binds to the promoter region of the *FvHsfB1a* gene (Figure 6). It has been reported that the WRKY TFs play important roles in abiotic stress responses, especially WRKY75 [52,53,54]. For example, the overexpression of *PtrWRKY75* in poplar and *AhWRKY75* in peanuts improves heat, drought, and salt tolerance; the overexpression of *MdWRKY75* in apple regulates the expression of *Hsf* genes to enhance thermotolerance [55,56,57]. In a previous study, we cloned the *FvWRKY75* gene and found that the *FvWRKY75* gene plays an active role in response to salt stress [34]. Herein, we explored the role of the *FvWRKY75* gene in the response to high-temperature stress. It was noticed that the *FvWRKY75*-OE transgenic *Arabidopsis* plants exhibited a better phenotype compared to the WT under heat treatment, and lower MDA and O_2_^−^ contents, higher chlorophyll and proline contents, and better SOD and POD activities were observed in the former than in the latter (Figure 7). This suggests that *FvWRKY75* is also a positive regulator of thermotolerance in strawberry.

In conclusion, we suggest a model of the WRKY-HSF transcriptional regulatory cascade for regulating heat stress in strawberry. In response to heat stress, *FvHsfB1a* acts as a positive regulator to improve thermotolerance by modulating ROS scavenging, enhancing antioxidant enzyme activity, and being directly regulated by the FvWRKY75 TF. These results provide physiological and molecular evidence highlighting the significance of *FvHsfB1a* in plant responses to heat stress.

## 4. Materials and Methods

### 4.1. Plant Materials, Growth Conditions, and Stress Treatment

Diploid strawberry cultivar ‘Ruegen’ (*Fragia vesca*), Nc89 tobacco (*Nicotiana tabacum*), and *Arabidopsis thaliana* (Columbia) plants were grown in a growth chamber at 23 °C, a 12 h light (4000 lux)/12 h dark photoperiod, and 60% relative humidity in Anhui Agricultural University (Hefei, China). The leaf, stem, root, flower, green fruit, and mature fruit of ‘Ruegen’ plants were harvested and used to detect the tissue specificity of the *FvHsfB1a* gene. The two-month-old seedings were subjected to 42 °C treatment in an artificial climate incubator, and mature leaves were collected at a series of time points (0 h, 3 h, 6 h, 12 h, 24 h, and 48 h).

### 4.2. Structure and Sequence Analysis of FvHsfB1a

The CDS sequence of *FvHsfB1a* (XM_004288037.1) was collected from NCBI (www.ncbi.nlm.nih.gov, accessed on 10 April 2022) and cloned from ‘Ruegen’ cDNA using the specific primers (Appendix A). DANMAN (version 6.0) software was used to analyze the homologs of FvHsfB1a and other species (accessed on 20 May 2022). MEGA (version 7.0) software was used to construct the phylogenetic tree by the maximum-likelihood (ML) method (accessed on 20 May 2022).

### 4.3. Subcellular Localization of FvHsfB1a Protein

The CDS region without the stop codon of *FvHsfB1a* was cloned and used to construct the fusion vector 35S::FvHsfB1a-eGFPm and then introduced into the *A. tumefaciens* GV3101 strain to infiltrate the tobacco leaves. Leica fluorescence confocal microscopy (Leica TCS SP8) was used to observe the GFP signal. The subcellular localization assays were used according to the methods of a previous study [58]. The primers used are listed in Appendix A.

### 4.4. Genetic Transformation of Arabidopsis

The CDS sequence of *FvHsfB1a* was cloned and inserted into the pRI101-AN vector to construct the pRI101-FvHsfB1a fusion vector. The fusion vector was transferred into the *A. tumefaciens* GV3101 strain and then transferred into the *Arabidopsis* Columbia-0 (Col-0, abbreviated WT) plants at flower budding by the floral dip method [59]. T3 homozygous transgenic lines were measured by PCR and qRT-PCR with the specific primers, and used for further tests. All of the primers used are listed in Appendix A.

### 4.5. RNA Extraction and RT-qPCR

The CTAB methods were used to isolate the total RNA of strawberry and *Arabidopsis* plants. Each cDNA sample was reverse-transcribed and then used for quantitative real-time PCR (qRT-PCR) according to the established protocols. Three technical replicates of each sample were analyzed for qRT-PCR. The primers used are listed in Appendix A.

### 4.6. Heat Stress Assay of Transgenic Arabidopsis Plants

To test the function of *FvHsfB1a* and *FvWRKY75* genes, WT, *FvHsfB1a*-OE, and *FvWRKY75*-OE lines were treated with heat stress. At first, seeds of WT and *FvHsfB1a*-OE lines were treated in a chamber at 37 °C and 42 °C in darkness for 7 days for heat stress. After heat treatment, all Arabidopsis seeds were transferred into a growth chamber for 4 days under normal growing conditions (23/18 °C day/night), and then the germination rates were measured. Meanwhile, seeds of *FvHsfB1a*-OE lines and WT were sterilized and seeded on an MS medium plate. A sample of 7-day-old Arabidopsis seedlings was subjected to heat stress at 45 °C for 3 h, and then underwent recovery for 5 days under normal conditions to record phenotypes and count survival rates. Furthermore, 3-week-old WT, *FvHsfB1a*-OE, and *FvWRKY75*-OE lines were subjected to heat stress at 42/37 °C (day/night) for 7 days, and samples were collected to measure the physiological parameters. A sample of 3-week-old plants was subjected to heat stress at 42 °C for 3 h, and the leaves were harvested to analyze the expression profiles of related genes.

### 4.7. Yeast One-Hybrid Assay

The CDS region of *FvWRKY75* (XM_004310052.2) was cloned and inserted into the plasmid vector pGADT7 with the GAL4 activation domain. PlantCARE online software (http://bioinformatics.psb.ugent.be/webtools/plantcare/html/, accessed on 20 May 2022) was used to identify the cis-acting elements in the promoter region. The promoter region (1500 bp) of *FvHsfB1a* was cloned and inserted into the pAbAi vector. Yeast one-hybrid assays were conducted according to the previously reported methods [58]. The primers used for the yeast one-hybrid assays are listed in Appendix A.

### 4.8. Dual Luciferase Activity Assay

The CDS region of *FvWRKY75* was cloned and inserted into the pGreenII-62SK vector, named pGreenII-62SK-FvWRKY75, and defined as an effector. The promoter region (1500 bp) of *FvHsfB1a* was inserted into the pGreenII-0800-LUC vector and named pGreenII-0800-*FvHsfB1a*, which acts as a reporter. Dual luciferase activity assays were conducted according to the previously reported methods [58]. All of the primers used are listed in Appendix A.

### 4.9. Data Analysis

The data are the average of three repeated experiments and are expressed as the mean ± SD. The statistical analysis was determined by using SPSS system (version 20) software. Duncan’s multiple range test was used to identify significant differences among treatment means (* *p* < 0.05, ** *p* < 0.01, and *** *p* < 0.001).

## Figures and Tables

**Figure 1 plants-14-02392-f001:**
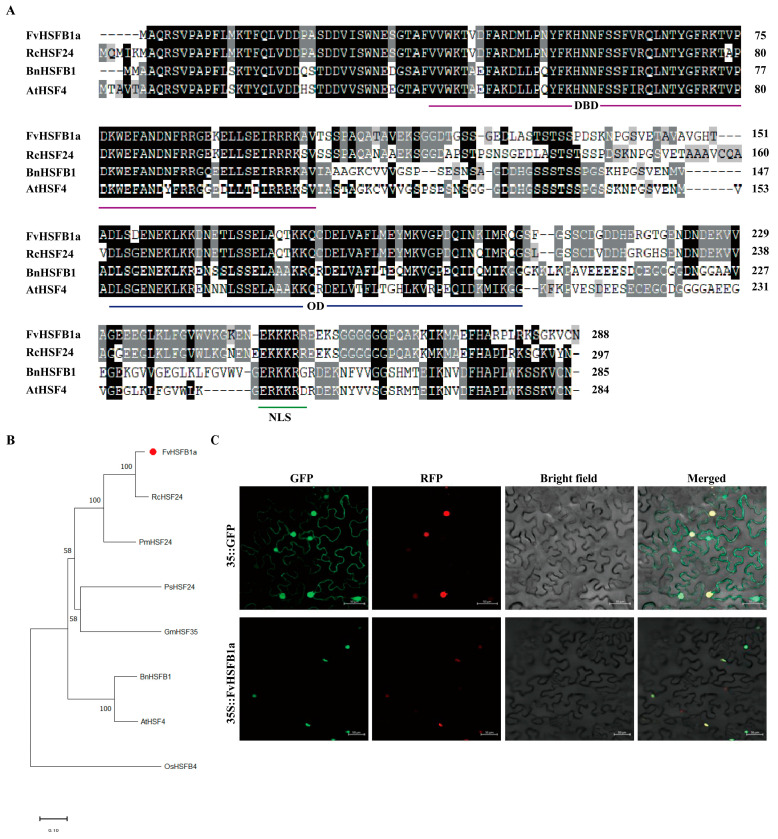
Analysis of FvHsfB1a transcription factor. (**A**) Amino acid sequence alignment of FvHsfB1a; DBD: DNA-binding domain; OD: oligomerization domain; NLS: nuclear localization signal of *Fragaria vesca*, *Rosa chinensis*, *Brassica napus*, and *Arabidopsis thaliana*. (**B**) Phylogenetic tree analysis of FvHsfB1a. (**C**) Subcellular localization of FvHsfB1a in tobacco leaves and RFP as a nuclear location marker.

**Figure 2 plants-14-02392-f002:**
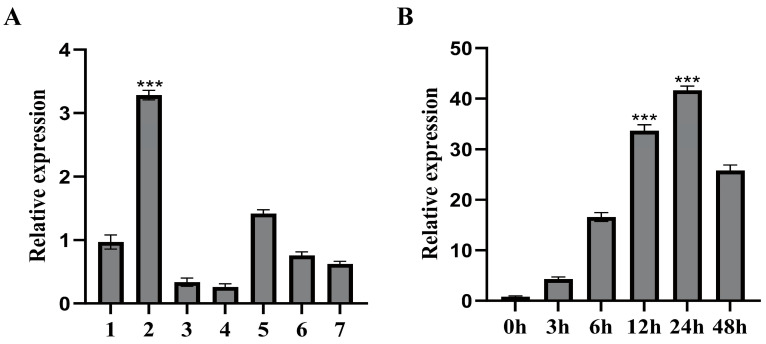
The expression profile of *FvHsfB1a* in diploid strawberry. (**A**) The tissue-specific expression of the *FvHsfB1a* gene in diploid strawberry. 1: Root; 2: stem; 3: young leaves; 4: mature leaves; 5: flowers; 6: young fruits; 7: mature fruits. (**B**) Relative expression levels of the *FvHsfB1a* gene under heat stress (42 °C). Three independent biological replicates were used for the experiment. *** *p* < 0.001.

**Figure 3 plants-14-02392-f003:**
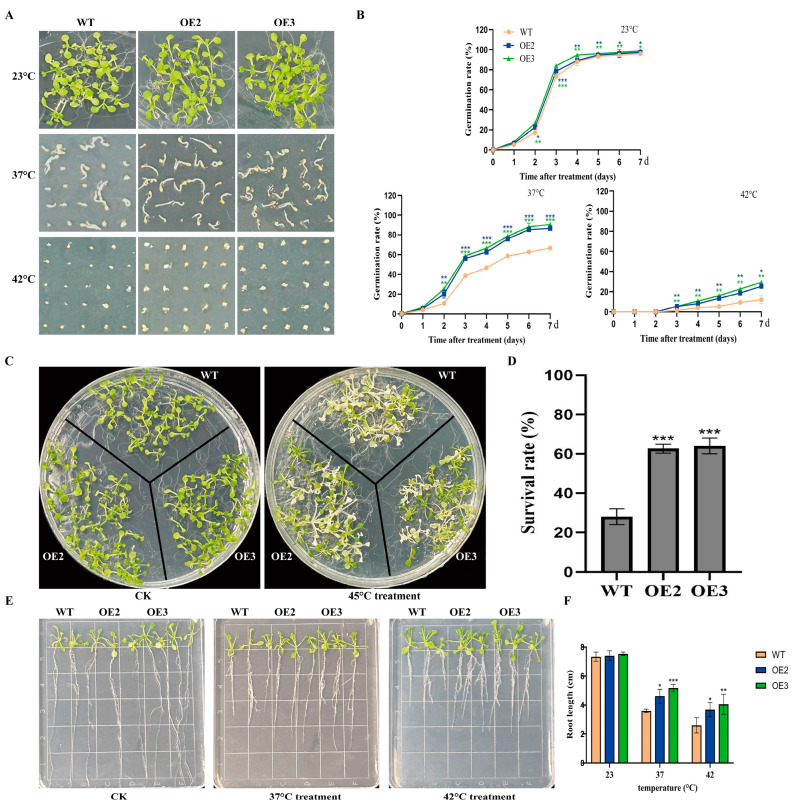
Heat tolerance analysis of WT and *FvHsfB1a*-OE lines. (**A**,**B**) Germination phenotypes and germination rates of *FvHsfB1a*-OEs and WT seeds under heat stress conditions. (**C**,**D**) Phenotypes and survival rates of *FvHsfB1a*-OEs and WT seedlings under heat stress. A sample of 7 d old seedlings grown on 1/2 MS medium was treated at 45 °C for 4 h, and then underwent recovery for 7 d. (**E**,**F**) The phenotype and root length of WT and *FvHsfB1a*-OE seedlings before and after 37 °C and 42 °C treatments, respectively. Three independent biological replicates were used for the experiment. *** *p* < 0.001, ** *p* < 0.01, * *p* < 0.05.

**Figure 4 plants-14-02392-f004:**
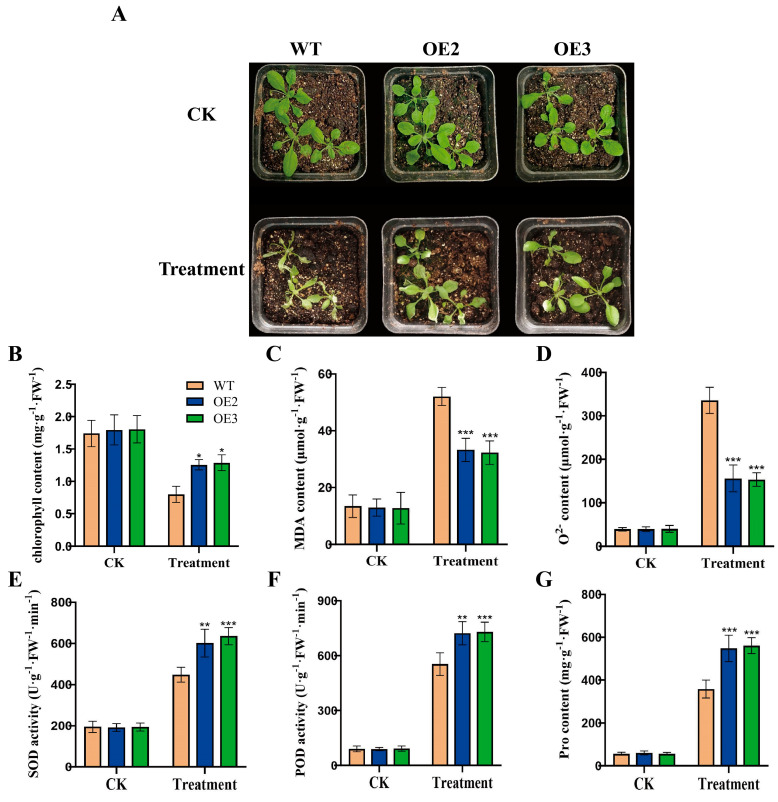
Thermotolerance analysis of WT and *FvHsfB1a*-OE lines. (**A**) Phenotypes of *FvHsfB1a*-OEs and WT plants before and after heat stress treatment for 7 d. (**B**–**G**) The chlorophyll, MDA, O^2−^, SOD activity, POD activity, and proline content of WT and *FvHsfB1a*-OE plants under heat stress, respectively. Three independent biological replicates were used for the experiment. *** *p* < 0.001, ** *p* < 0.01, * *p* < 0.05.

**Figure 5 plants-14-02392-f005:**
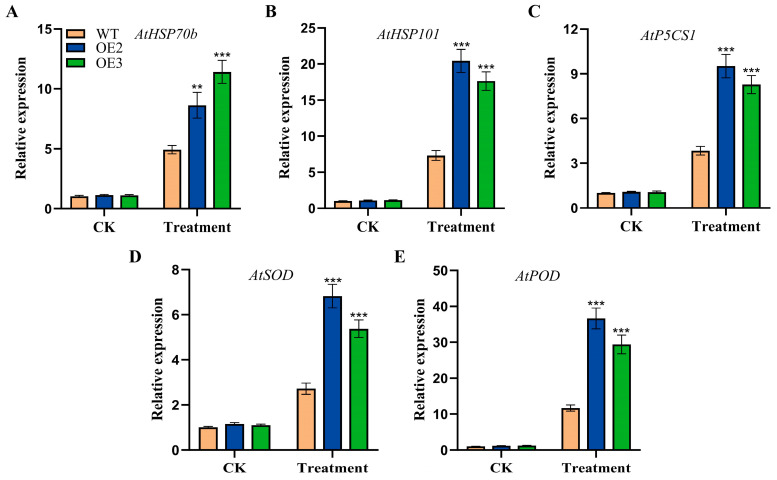
The expression of stress-related genes in WT and *FvHsfB1a*-OE lines under cold stress. The gene expression levels of 2-week-old *Arabidopsis* seedlings treated at 42 °C for 2 d were detected by using qRT-PCR. (**A**–**E**) The transcript level of *AtHSP70b*, *AtHSP101*, *AtP5CS1*, *AtSOD* and *AtPOD* gene in mature plants under heat stress, respectively. Three independent biological replicates were used for the experiment. *** *p* < 0.001, ** *p* < 0.01.

**Figure 6 plants-14-02392-f006:**
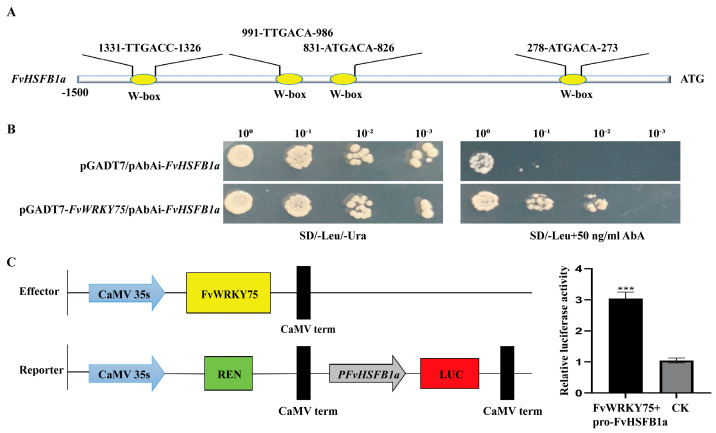
FvWRKY75 activated *FvHsfB1a* expression. (**A**) Bioinformatics analysis showed that the promoter of *FvHsfB1a* contained WRKY binding elements. (**B**) Analysis of the activation of *FvHsfB1a* transcription by FvWRKY75 using a yeast one-hybrid assay. (**C**) The luciferase activity assay showed that FvWRKY75 binds to the *FvHsfB1a* promoter. *** *p* < 0.001.

**Figure 7 plants-14-02392-f007:**
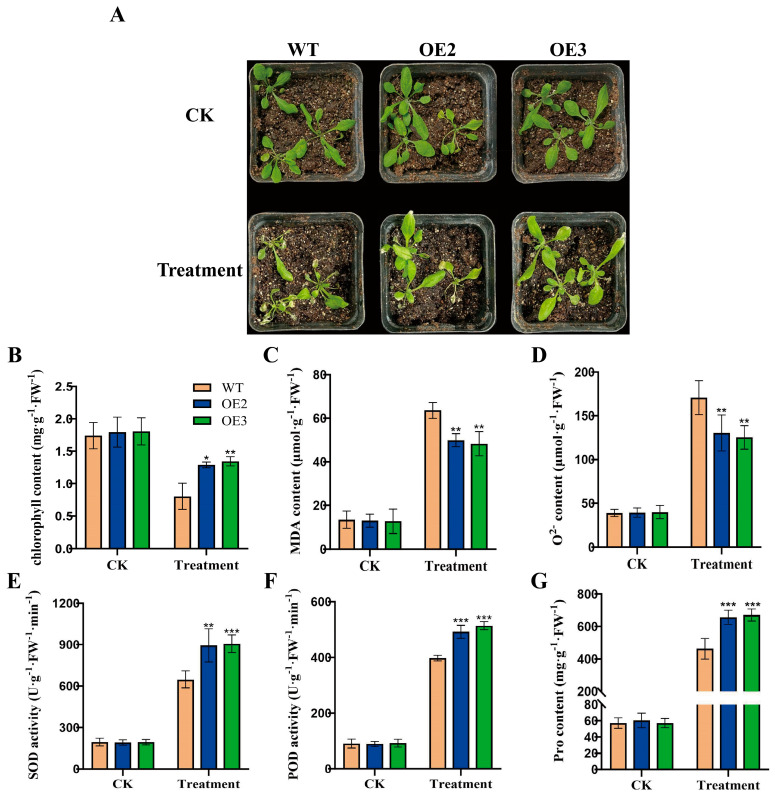
Heat tolerance analyses of WT and *FvWRKY75*-OE lines. (**A**) Phenotypes of *FvWRKY75*-OEs and WT plants with or without heat treatment for 7 d. (**B**–**G**) The chlorophyll, MDA, O^2−^, SOD activity, POD activity, and proline content of WT and *FvWRKY75*-OE plants under heat stress, respectively. Three independent biological replicates were used for the experiment. *** *p* < 0.001, ** *p* < 0.01, * *p* < 0.05.

## Data Availability

Data are contained within the article and can be made available on request.

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
