# Peer review of "FvHsfB1a Gene Improves Thermotolerance in Transgenic Arabidopsis"

_plants, 2025, doi:10.3390/plants14152392_

Round 1

Reviewer 1 Report

Comments and Suggestions for Authors

Overall comments

The MS entitled "FvHsfB1a gene improves the thermotolerance in transgenic Arabidopsis" seems to be well-designed and written. In my opinion, the MS should be accepted for publication after addressing few minor comments.

Specific comments

(a)Figure1 quality is poor, author should improve the quality of figure.

(b)Figure legends should be self-explanatory; duration of heat treatment time should be included in the legend of figure 4.

(c)As shown in figure 4, why the phenotype of over-expressed lines is poor as compared to wild type plants even under control conditions.

Author Response

Comments 1: Figure1 quality is poor, author should improve the quality of figure.

Response 1: Thank you for pointing this out. I agree with this comment. I provided a new Fig. 1 with  higher quality.

Comments 2: Figure legends should be self-explanatory; duration of heat treatment time should be included in the legend of figure 4.

Response 2: Thank you for pointing this out. I agree with this comment. I provided the heat treatment time (7d) in the legend of figure 4.

Comments 3: As shown in figure 4, why the phenotype of over-expressed lines is poor as compared to wild type plants even under control conditions.

Response 3: Thank you for pointing this out. I agree with this comment. This is my mistake. I will add a new picture of Fig.4A with better phenotype of FvHsfB1a-OE lines in this article.

Reviewer 2 Report

Comments and Suggestions for Authors

Comments and Suggestions for Authors: plants-3753632

Dear authors,

1. In general, Hsfs are a family of transcription factors that are highly conserved across plant species, and they are typically classified into three classes: HSFA, HSFB, and HSFC. In wild diploid woodland strawberries, there might be many FvHsf family genes. Why did the author choose FvHsfB1a for investigation?

2. For confirmation of Arabidopsis transgenic lines overexpressing FvHsfB1a, the authors used RT-PCR and qRT-PCR analyses (L257-258, L97-98, and result shown in Supplementary Figure S1). I wonder why the author did not use genomic DNA PCR, which is known to be a more common and convenient method to confirm foreign genes in transgenic plants, since Arabidopsis transgenic plants ectopically overexpressing the FvHsfB1a gene of strawberries.

3. Why did the authors select the different lengths of promoter region of FvHsfB1a for Yeast One-Hybrid Assay and Dual Luciferase Activity Assay? For instance, herein, 1400 bp (L281) and 1500 bp (L287) of the FvHsfB1a promoter region were cloned into the pAbAi and pGreenII-0800-LUC vector for Yeast One-Hybrid Assay and Dual Luciferase Activity Assay, respectively.

4. What bioinformatics analysis was used for finding out the promoter of the FvHsfB1a gene containing WRKY binding elements (relative data, Fig.6A). This should be described in the Materials and Methods

5. Note that I could not access the primers used in this study, which authors listed in Table S1. Please include it in the Supplementary Materials. In addition, provide the Accession number/Genbank of genes: FvHsfB1a and FvWRKY75, which are based on their CDS sequences (from NCBI) that the authors cloned into the expression vectors, then made transgenic plants expressing these genes.

6. Fig.1 A-B shows amino acid sequence alignment and phylogenetic tree analysis of FvHsfB1a. However, what/which other species (L243) were used for homology and the phylogenetic tree analysis was not indicated. Please clarify. Additionally, the notation in Fig. 1A, such as NLS, DBD, and OD, should also be noted in Fig.1’s legend. For example, DBD = DNA-binding domain, OD = oligomerization domain.

7. The legend of Fig.2 needs to be revised to the same as the notation of the x-axis. The legend of the x-axis was 1, 2…7. However, the legend of Fig.2 was presented as “A: root; B: stem; C: young leaves; D: mature leaf; E: flowers; F: young fruit; G: mature fruit.” (L92-93).

8. Some figures (Fig.1, Fig.3B) lack clarity. Please enhance the resolution.

9. Remarks:

  • The full form of these abbreviations should be presented when it is first mentioned: MDA (L22), SOD and POD (L109). Moreover, the full names of stress-related genes (AtP5CS1, AtSOD, and AtPOD) (L129) should be presented.
  • Revise these words for more proper meaning: “fertilize” (L249) → “infiltrate”; “fertilized” (L567) → “transformed into”

I have highlighted the above comments and remarks on the manuscript. Please use it for easy tracking and revision.

With regards

Author Response

Comments 1:  In general, Hsfs are a family of transcription factors that are highly conserved across plant species, and they are typically classified into three classes: HSFA, HSFB, and HSFC. In wild diploid woodland strawberries, there might be many FvHsf family genes. Why did the author choose FvHsfB1a for investigation?

Response 1: Thank you for pointing this out. Many FvHsf family genes were identified in diploid woodland strawberry (Fragaria vesca). In this article, we found that the FvHsfB1a gene was highly induced by heat stress. Previous studies reported that HsfBs genes may played diverse roles in response to heat stress. We wanted to explore the function of FvHsfB1a gene in response to heat stress. In addition, we also explored the function of several FvHsfs genes and prepared for submission in other journals.

Comments 2: For confirmation of Arabidopsis transgenic lines overexpressing FvHsfB1a, the authors used RT-PCR and qRT-PCR analyses (L257-258, L97-98, and result shown in Supplementary Figure S1). I wonder why the author did not use genomic DNA PCR, which is known to be a more common and convenient method to confirm foreign genes in transgenic plants, since Arabidopsis transgenic plants ectopically overexpressing the FvHsfB1a gene of strawberries.

Response 2: Thank you for pointing this out. In this study, we used genomic DNA to verify whether the FvHsfB1a transgenic Arabidopsis plants were generated by RT-PCR (Figure S1A).

Comments 3: Why did the authors select the different lengths of promoter region of FvHsfB1a for Yeast One-Hybrid Assay and Dual Luciferase Activity Assay? For instance, herein, 1400 bp (L281) and 1500 bp (L287) of the FvHsfB1a promoter region were cloned into the pAbAi and pGreenII-0800-LUC vector for Yeast One-Hybrid Assay and Dual Luciferase Activity Assay, respectively.

Response 3: Thank you for pointing this out. We select 1500 bp promoter region of FvHsfB1a for Yeast One-Hybrid Assay and Dual Luciferase Activity Assay. Here, we made a mistake in write 1400bp.

Comments 4: What bioinformatics analysis was used for finding out the promoter of the FvHsfB1a gene containing WRKY binding elements (relative data, Fig.6A). This should be described in the Materials and Methods.

Response 4: Thank you for pointing this out. We used Plantcare to analysis the promoter of FvHsfB1a gene, and added it in the materials.

Comments 5: Note that I could not access the primers used in this study, which authors listed in Table S1. Please include it in the Supplementary Materials. In addition, provide the Accession number/Genbank of genes: FvHsfB1a and FvWRKY75, which are based on their CDS sequences (from NCBI) that the authors cloned into the expression vectors, then made transgenic plants expressing these genes.

Response 5: Thank you for pointing this out. I am deeply sorry for this. I will add the Table S1 and accession number in this article.

Comments 6: Fig.1 A-B shows amino acid sequence alignment and phylogenetic tree analysis of FvHsfB1a. However, what/which other species (L243) were used for homology and the phylogenetic tree analysis was not indicated. Please clarify. Additionally, the notation in Fig. 1A, such as NLS, DBD, and OD, should also be noted in Fig.1’s legend. For example, DBD = DNA-binding domain, OD = oligomerization domain.

Response 6: Thank you for pointing this out. I will add the different plant species and explain the meaning of NLS, DBD, and OD on the Fig.1 legend.

Comments 7: The legend of Fig.2 needs to be revised to the same as the notation of the x-axis. The legend of the x-axis was 1, 2…7. However, the legend of Fig.2 was presented as “A: root; B: stem; C: young leaves; D: mature leaf; E: flowers; F: young fruit; G: mature fruit.” (L92-93).

Response 7: Thank you for pointing this out. This is my mistake. I will modify the notation of the x-axis on the legend of Fig.2

Comments 8: Some figures (Fig.1, Fig.3B) lack clarity. Please enhance the resolution. 

Response 8: Thank you for pointing this out. I added a new Fig.1 with higher definition in this article. But I don 't think the clarity of Fig. 3B is poor.

Comments 9:Remarks: The full form of these abbreviations should be presented when it is first mentioned: MDA (L22), SOD and POD (L109). Moreover, the full names of stress-related genes (AtP5CS1, AtSOD, and AtPOD) (L129) should be presented.

Revise these words for more proper meaning: “fertilize” (L249) → “infiltrate”; “fertilized” (L567) → “transformed into”

Response 9: Thank you for pointing this out. I will modify those mistake in this article. 

nuclear localization signal

Reviewer 3 Report

Comments and Suggestions for Authors

In this paper, the authors characterize the role of the woodland strawberry FvHsfB1a protein in thermotolerance. They present the sequence analysis of FvHsfB1a ana data on its subcellular localization. Further, the authors show that transgenic Arabidopsis lines overexpressing FvHsfB1a show an increased thermotolerance. An important part of the work is devoted to gene expression regulation involving FvHsfB1a. First, FvHsfB1a is shown to affect the expression of a number of stress-related genes. Second, WRKY75 is found to activate the FvHsfB1a promoter. Moreover, overexpression of WRKY75 in transgenic Arabidopsis is shown to increase thermotolerance. In my opinion, this paper can be published in Plants; however, I have concerns that should be addressed before the paper is accepted for publication.

Major points

  1. The experiment on the subcellular localization of FvHsfB1a is not adequately described in Section 2.1 nor properlypresented in Figure 1C. Additionally, there is a contradiction between the text and the figure. Line 77 states that FvHsfB1a is fused with GFP; however, Figure 1C shows the co-localization of FvHsfB1a fused to RFP with non-fused GFP, as I understand it. Thus, this experiment should be presented properly.
  2. Lines 98-99. When introducing the FvHsfB1a-overexpressing lines (OE lines), the authors state that “no significant difference in the seed germination and seedlings growth between WT and OE lines was observed under normal conditions”. However, as Figure 4A clearly shows, the OE lines have a different phenotype than the WT plants. The leaves of the OE lines appear distorted and exhibit unusual coloration and probably even spontaneous necrosis. This OE line phenotype should be presented in higher-quality images and thoroughly described in the text. Additionally, the text should discuss the tradeoff between constitutive activation of FvHsfB1a and the plant's ability to maintain a normal phenotype and vegetative characteristics. It would also be interesting to determine which FvHsfB1-dependent defense pathways are responsible for this phenotype. This point can be added to the discussion.

Minor points

Abstract: The first two sentences seem to have very similar meanings.

Line 10: HsfBs is not defined. Besides, it seems that the authors mean Hsfs here.

Lines 11-12: something is missing in this sentence.

Line 18: ROS is not defined

Line 22: MDA is not defined

Lines 28-35. I suggest removing the first paragraph of the Introduction, as this information is too general.

Line 60. What is Ruegen? – Should be defined.

Line 73: Why the isoelectric point is measured in moles (M)?

Lines 86-88. Which tissues and organs were analyzed for FvHsfB1a expression profiles under various stress conditions?

Line 132: cold stress should be heat stress.

Line 148: 2 should be two

There are also issues with the English. Below are some examples. In addition to addressing these specific issues, I strongly recommend that the entire text be checked by a native speaker or a language service.

Line 13: at N terminus

Line 17: thermotolerance within the higher germination rate and survival rate

Line 19: better enzyme activities

Line 20: FvWRKY75 activate

Line 85: was differential expressed

Line 110: could improves

Line 151: better phenotypic

Author Response

Comments 1: The experiment on the subcellular localization of FvHsfB1a is not adequately described in Section 2.1 nor properlypresented in Figure 1C. Additionally, there is a contradiction between the text and the figure. Line 77 states that FvHsfB1a is fused with GFP; however, Figure 1C shows the co-localization of FvHsfB1a fused to RFP with non-fused GFP, as I understand it. Thus, this experiment should be presented properly.

Response 1: Thank you for pointing this out. I agree with this comment. In this article, RFP as a nuclear localization marker. In Figure 1C, we can observed the GFP and RFP signal of FvHsfB1a in nucleus, which indicated that protein was only localized in nucleus. 

Comments 2: Lines 98-99. When introducing the FvHsfB1a-overexpressing lines (OE lines), the authors state that “no significant difference in the seed germination and seedlings growth between WT and OE lines was observed under normal conditions”. However, as Figure 4A clearly shows, the OE lines have a different phenotype than the WT plants. The leaves of the OE lines appear distorted and exhibit unusual coloration and probably even spontaneous necrosis. This OE line phenotype should be presented in higher-quality images and thoroughly described in the text. Additionally, the text should discuss the tradeoff between constitutive activation of FvHsfB1a and the plant's ability to maintain a normal phenotype and vegetative characteristics. It would also be interesting to determine which FvHsfB1-dependent defense pathways are responsible for this phenotype. This point can be added to the discussion.

Response 2: Thank you for pointing this out. I agree with this comment. I will add a new picture of Fig.4A with higher quality in this article.

Comments 3: Minor points

Abstract: The first two sentences seem to have very similar meanings.

Line 10: HsfBs is not defined. Besides, it seems that the authors mean Hsfs here.

Lines 11-12: something is missing in this sentence.

Line 18: ROS is not defined

Line 22: MDA is not defined

Lines 28-35. I suggest removing the first paragraph of the Introduction, as this information is too general.

Line 60. What is Ruegen? – Should be defined.

Line 73: Why the isoelectric point is measured in moles (M)?

Lines 86-88. Which tissues and organs were analyzed for FvHsfB1a expression profiles under various stress conditions?

Line 132: cold stress should be heat stress.

Line 148: 2 should be two

Response 3: Thank you for pointing this out. I will modify those mistake according to the expert opinion of the reviewer in this article.

Comments 4: There are also issues with the English. Below are some examples. In addition to addressing these specific issues, I strongly recommend that the entire text be checked by a native speaker or a language service.

Line 13: at N terminus

Line 17: thermotolerance within the higher germination rate and survival rate

Line 19: better enzyme activities

Line 20: FvWRKY75 activate

Line 85: was differential expressed

Line 110: could improves

Line 151: better phenotypic

Response 4: Thank you for pointing this out. The entire text will be checked by a language service. I will provide a polished article.

Round 2

Reviewer 2 Report

Comments and Suggestions for Authors

Dear authors,

The authors have made efforts to revise the manuscript. However, some of my comments, which I have raised in Round 1, need to be revised further. I would like to bring here my comments and the authors' responses:

Comment 2:   For confirmation of Arabidopsis transgenic lines overexpressing FvHsfB1a, the authors used RT-PCR and qRT-PCR analyses (L257-258, L97-98, and result shown in Supplementary Figure S1). I wonder why the author did not use genomic DNA PCR, which is known to be a more common and convenient method to confirm foreign genes in transgenic plants, since Arabidopsis transgenic plants ectopically overexpressing the FvHsfB1a gene of strawberries.

Response 2: Thank you for pointing this out. In this study, we used genomic DNA to verify whether the FvHsfB1a transgenic Arabidopsis plants were generated by RT-PCR (Figure S1A).

Re-comment 2: I do not agree with the authors’ response on this point. RT-PCR relies on reverse transcriptase, which synthesizes DNA from an RNA template. Therefore, RT-PCR requires RNA as a template to create complementary DNA (cDNA), which is then amplified by PCR. While genomic DNA PCR uses (genomic) DNA as a template for regular PCR, it cannot be directly used in RT-PCR. For confirmation or detection of the foreign gene ectopically expressed in transgenic plants, normal PCR (gDNA PCR) is commonly used. Please review and revise the main text of the manuscript.

Comment 5: Note that I could not access the primers used in this study, which authors listed in Table S1. Please include it in the Supplementary Materials. In addition, provide the Accession number/Genbank of genes: FvHsfB1a and FvWRKY75, which are based on their CDS sequences (from NCBI) that the authors cloned into the expression vectors, then made transgenic plants expressing these genes.

Response 5: Thank you for pointing this out. I am deeply sorry for this. I will add the Table S1 and accession number in this article.

Re-comment 5: Thank you for including Table S1 in Supplementary Materials. Other than FvHsfB1a (XM_004288037.1) and FvWRKY75 (XM_004310052.2, please also provide the accession number/GenBank of genes listed in Table S1, based on which the primer sequences were designed.

New comment: The manuscript lacks of Conclusions section

With regards,

Author Response

Re-comment 2: I do not agree with the authors’ response on this point. RT-PCR relies on reverse transcriptase, which synthesizes DNA from an RNA template. Therefore, RT-PCR requires RNA as a template to create complementary DNA (cDNA), which is then amplified by PCR. While genomic DNA PCR uses (genomic) DNA as a template for regular PCR, it cannot be directly used in RT-PCR. For confirmation or detection of the foreign gene ectopically expressed in transgenic plants, normal PCR (gDNA PCR) is commonly used. Please review and revise the main text of the manuscript.

Response 2: Thank you for pointing this out. I will modify this mistake in this article (use PCR to verify the FvHsfB1a transgenic Arabidopsis).

Re-comment 5: Thank you for including Table S1 in Supplementary Materials. Other than FvHsfB1a (XM_004288037.1) and FvWRKY75 (XM_004310052.2, please also provide the accession number/GenBank of genes listed in Table S1, based on which the primer sequences were designed.

Response 5: Thank you for pointing this out. I will add the accession number/GenBank of genes listed in Table S1.

New comment: The manuscript lacks of Conclusions section.

Response: Thank you for pointing this out. At the end of the discussion, we provide a part of conclusion.